# Dense Phases of γ-Gliadins in Confined Geometries

**Amélie Banc** [1], **Laurence Navailles** [2], **Jacques Leng** [3] and **Denis Renard** [4,*]

[1]  CNRS, Laboratoire Charles Coulomb (L2C), Université de Montpellier, 34095 Montpellier, France; amelie.banc@umontpellier.fr

[2]  CNRS, Centre de recherche Paul-Pascal, 115 Avenue du Docteur Schweitzer, 33600 Pessac, France; laurence.navailles@crpp-bordeaux.cnrs.fr

[3]  Laboratoire du Futur, Université Bordeaux-1, 178 Avenue du Docteur Schweitzer, CEDEX, 33608 Pessac, France; jacques.leng-exterieur@solvay.com

[4]  INRAE, BIA Biopolymères Interactions Assemblages, BP 71627, CEDEX 03, 44316 Nantes, France

*  Correspondence: denis.renard@inrae.fr; Tel.: +33-240-67-5052

**Abstract:** The binary phase diagram of γ-gliadin, a wheat storage protein, in water was explored thanks to the microevaporator, an original PDMS microfluidic device. This protein, usually qualified as insoluble in aqueous environments, displayed a partial solubility in water. Two liquid phases, a very dilute and a dense phase, were identified after a few hours of accumulation time in the microevaporator. This liquid–liquid phase separation (LLPS) was further characterized through in situ micro-Raman spectroscopy of the dilute and dense protein phases. Micro-Raman spectroscopy showed a specific orientation of phenylalanine residues perpendicular to the PDMS surfaces only for the diluted phase. This orientation was ascribed to the protein adsorption at interfaces, which would act as nuclei for the growth of dense phase in bulk. This study, thanks to the use of both aqueous solvent and a microevaporator, would provide some evidence for a possible physicochemical origin of the gliadin assembly in the endoplasmic reticulum of albumen cells, leading to the formation of dense phases called protein bodies. The microfluidic tool could be used also in food science to probe protein–protein interactions in order to build up phase diagrams.

**Keywords:** gliadin; icroevaporation; protein bodies; liquid–liquid phase separation

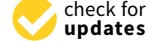

## 1. Introduction

Wheat is one of the most important resources of human food, in part due to the unique properties of gluten. Indeed, the latter is made of wheat storage proteins that produce upon hydration a unique basis, a sticky and cohesive paste, for processing a large variety of foodstuff. From a technological point of view, the behavior of such dough depends tremendously on the process via mechanical shearing, added salt, heat, etc., and a thorough link between the exact composition of wheat storage proteins and technological issues remains to be established [1]. These proteins also seem promising in terms of biomaterials and drug delivery systems [2–4].

From a biological point of view, these proteins are required as a source of carbon, sulfur, and nitrogen during seed germination. Localized in the endosperm of the grain, these proteins are composed of two main subgroups, gliadins and glutenins [5]. At the early stages of grain development, these proteins are stored in protein bodies, which are micron-sized organelles, containing up to 80% of proteins, and that originate from the endoplasmic reticulum [6]. The secondary and tertiary structures of wheat storage proteins are still ill defined [7–11]. The in vivo quaternary structure of wheat proteins, which should result from their assembly during the protein bodies genesis, is even more puzzling. Biologists show that two distinct secretory pathways generate these bodies: the classical pathway via the Golgi apparatus and a pathway by which protein bodies emerge directly from the endoplasmic reticulum bypassing the Golgi apparatus [12,13]. The mechanisms implied in the latter route, which concern only plant species for the storage of proteins, oils,

and enzymes, are still elusive. These mechanisms constitute an important question for the plant biology [14–17]. It was shown that an S-rich gliadin, γ-gliadin could accumulate using this route following their expression in various systems (yeast [18], xenopus oocytes [19,20], tobacco [21] cells) such as in wheat endosperm cells. This observation suggested that wheat storage proteins, such as γ-gliadin, contain sufficient information to initiate the formation of protein bodies in the endoplasmic reticulum of heterologous systems.

Hence, mechanisms mainly based on physicochemical properties of such proteins can be proposed. Gliadins were found to form in vitro condensates in 55% ethanol/water mixture by decreasing temperature [22]. The possible role of this liquid–liquid phase separation (LLPS) process on the in vivo gliadin storage is elusive and remains to be explored. Bioinformatics analyses suggest that γ-gliadin is a hybrid protein with N-terminal domain predicted to be disordered and C-terminal domain predicted to be ordered. Spectroscopic data highlight the disordered nature of γ-gliadin. The ability of γ-gliadin to self-assemble into dynamic droplets through LLPS has been shown [10]. The importance of the predicted ordered C-terminal domain of γ-gliadin in the LLPS has also been demonstrated by highlighting the protein condensates transition from a liquid to a solid state under reducing conditions. Using a combination of computational, biochemical, and biophysical tools, researchers in a study of each of its N and C terminal domain revealed that γ-gliadin is a partially disordered protein with an unfolded N-terminal domain surprisingly resistant to chymotrypsin and a folded C-terminal domain [23]. As these proteins are very poorly soluble in aqueous solutions [24], the whole process could be 'surface-assisted' by the membranes of the endoplasmic reticulum while the confinement would enhance aggregation [25].

It is known that gliadins display an interfacial activity, reducing the air/water surface tension [26–28] and adsorbing onto different substrates [29]. Our previous studies [30,31] confirm this adsorption at both air–water and membrane–water interfaces, showing that γ-gliadins self-assembled into macromolecular aggregates, probably through intermolecular β-sheets, that grew from the interfaces side. More recently, a macromolecular assembly of wheat proteins within osmosomes (giant unilamellar vesicles with α-hemolysin pores) by pH-triggered LLPS within a few minutes was shown [32]. However, the role of membranes present in osmosome was not directly proven to be involved in the LLPS phenomenon. These studies suggested a possible route for surface mediated assembly of γ-gliadins. In this work, we extended this study to bulk assembly of confined γ-gliadins in aqueous solution using a microfluidic device that permits concentration on protein solutions. The specificity of our device is twofold: it permits concentration at a controlled pace of any solute, regardless of its initial state. This is particularly relevant in terms of γ-gliadins, for which there is no consensus as to their exact solubility in water. Moreover, the microfluidic device naturally yields a confined geometry (typical thickness of 10 μm) where surface effects are of prime importance. With this device, we were able to investigate the phase behavior of γ-gliadins in aqueous solution and to generate dense phases of these proteins. We observed that these phases nucleated first on a surface and further grew in bulk. Raman microspectroscopy revealed a preferential orientation of some residues close to the surface and no significant orientation in bulk.

## 2. Materials and Methods

### 2.1. Solutions of γ-Gliadins

γ-Gliadins were prepared by extraction of gliadins from defatted gluten in ethanol/water mixtures followed by several steps of chromatographic purification as detailed elsewhere [31]. Proteins were labeled with a fluorescent dye (tetramethylrhodamine isothiocyanate, TRITC) using the procedure described previously [30]. The remaining free fluorescent label was removed by extensive dialysis against an ethanol/water mixture, and the protein was then freeze-dried. γ-gliadins are poorly soluble in water; we, however, prepared diluted aqueous solutions to concentrate them into the microfluidic device in a controlled manner. We mixed a small quantity of γ-gliadin powder (about 1 mg/mL)

in pure deionized water and kept the solution under constant magnetic stirring for 24 h. We then collected the supernatant after 30 min of centrifugation at 1000 rpm to remove the undissolved material. UV-visible spectroscopy permitted the measurement of a typical protein concentration corresponding to 10% of the initial concentration (i.e., 0.1 mg/mL). If necessary, the solution was further diluted with deionized water. These solutions were optically transparent, and dynamic light scattering revealed that they were mainly composed of γ-gliadin monomers characterized by a hydrodynamic radius ($R_h$) of 4 nm.

### 2.2. Microfluidic Device and Kinetic Inspection of Phase Diagram

A microfluidic device that permits concentrating any solution or dispersion, from a very diluted state up to a concentrated phase at the nanoliter scale [33,34], is briefly described below. The device works by extraction of solvent, which in turn concentrates the solute. The device used in this paper is a two-layer polydimethylsiloxane (PDMS) on glass microsystems (Figure 1A) (fabrication procedure detailed in [34]). The microchannels of the bottom layer are filled with the solution of interest, while air is circulated through the microchannel of the top layer to remove the water that pervaporates through the thin membrane of PDMS that separates the two networks (thickness *e* in the 10–30 μm range). A finger-like geometry (Figure 1A) characterized by a dead-end channel of rectangular cross-section (height *h*, width *w*, length *L*) is connected to a larger (millimetric) feeding open reservoir containing the solution to be concentrated. A terminal section of length $L_0 < L$ (typically millimeters to centimeters) is covered by the water removal network. The glass slide onto which the device is sealed is a 170 μm thick microscope slide, which allows high-resolution optical and confocal microscopy.

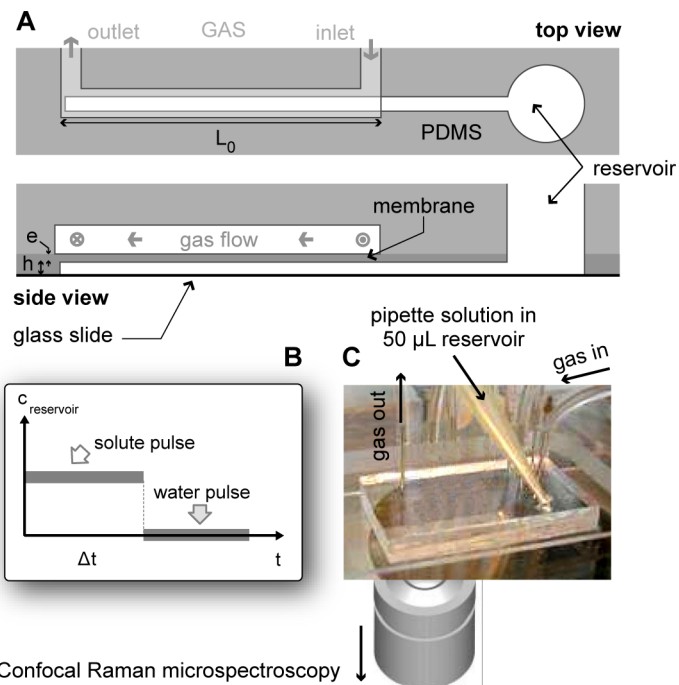

**Figure 1.** (**A**) Sketch of the microfluidic device used for concentrating γ-gliadin solutions: top and side views showing the gas and liquid microchannels and the thin PDMS membrane in between (typical dimensions: *e* = 10 μm, *h* = 25 μm, *w* = 200 μm, $L_0$ = 10 mm). (**B**) Flushing procedure used to confine solutes at the tip of the microchannel for micro-Raman analysis. (**C**) View of the microsystem being filled (or refilled) through the open access reservoir, and position of the confocal micro-Raman objective.

The operation principle of the device is the following: water in the bottom channel pervaporates through the thin membrane, which induces a compensating flow from the reservoir and concentration of the convected solutes at the tip of the finger if present in the

reservoir. The small dimensions lead to fast thermal regulation that permits isothermal studies, and direct observations of the induced phases and phenomena can be achieved thanks to the PDMS transparency.

The strength of the device is to permit the concentration at a controlled pace of any type of solute, even if very dilute in the reservoir. We showed that the concentration rate in the device could be rationalized on the basis of (i) its extraction efficiency characterized by an evaporation time $t_e \sim 100-1000$ s (depending for instance on the thickness of the extraction membrane), which represents the time needed to evaporate the volume of one channel and which can be calibrated precisely; (ii) the geometry and especially the evaporation length $L_0$; and (iii) a solute-dependent feature that accounts for the concentration mechanism [33,34].

More precisely, the latter occurs via a competition between the evaporation induced convection and the diffusion of the solute. Any solubilized species in water is thus convected and is trapped at the tip of the microdevice in an accumulation box of spatial extent $p = (Dt_e)^{1/2}$, where $D$ is the diffusion coefficient of the solute (here, $p \sim 70–200$ µm). On this basis, a rough yet realistic estimate of concentration rate of the solute at the tip of the microdevice is given by

$$\frac{\Delta c}{\Delta t} = \frac{L_0}{p}\frac{c_0}{t_e} \qquad (1)$$

where $c_0$ is the concentration of solute in the reservoir. The exact concentration $c(x,t)$ anywhere and at any time can be calculated exactly only in limited cases (e.g., ideal solutions) [34]. This estimate nevertheless serves a useful basis to extract information on the concentration process: after calibration ($t_e$), the waiting time between the start of the experiment and any observation can be translated into an equivalent concentration using Equation (1).

### 2.3. Microscopy and Spectroscopy

Most of the observations were carried out using standard microscopy (bright field, phase contrast or polarization). To study TRITC-labeled γ-gliadins, we performed fluorescence microscopy using a standard microscope (equipped with a mercury lamp and adequate filters). Raman confocal micro-spectroscopy was used to characterize in situ the dense phases of the unlabeled protein after concentration (Figure 1C). Confocal microscopy was actually required to collect the light scattered by the sample itself and not the surroundings. We used a bright ×100 water immersion objective (numerical aperture 1.4) mounted on a microscope and connected to a Jobin-Yvon spectrometer. This micro-spectroscopy device yields a depth of field in the $z$ direction $z \approx 10$–20 µm, while the lateral resolution is of the order of 1 µm. Raman scattering was collected at wavenumbers between 200 and 2000 cm$^{-1}$ after excitation at $\lambda = 785$ nm.

## 3. Results

### 3.1. Microscopic Observations of γ-Gliadin Accumulation in Pure Water

γ-Gliadins were dispersed in water as described previously. Steady accumulation experiments were carried out using the microfluidic device with dilute solutions ($\approx 0.1$ mg/mL) of bare and fluorescently labeled γ-gliadins, with both solutions always displaying the same behavior. After few hours of accumulation time, we observed the formation of droplets nucleated from the interfaces around the tip of the capillary (Figure 2A,B). This zone was enriched by the feeding mechanism of the device, and the more we waited, the more concentrated the zone was. By increasing accumulation time, the initial droplets, nucleated at interfaces, gradually detached to form spherical droplets with rough surfaces in the bulk of the capillary. When carrying this experiment with TRITC-labeled proteins, we clearly noticed that the protein solution phase separated (Figure 2A), segregating in dense liquid-like droplets and dispersed species responsible for a non-zero fluorescent background. It suggested a region of coexisting phases and thus underlying a liquid–liquid phase separation (LLPS). It is possible to deduce from the waiting time the concentration within the device [34]. Using the diffusion coefficient of gliadin monomers, we found that

the concentrations at the tip of the capillary was $200 \pm 5$ mg/mL, yielding a density of the dense phase of $1.05 \pm 0.05$ using an average protein density of 1.35 [35]. Interestingly, this liquid phase density is comparable with that estimated in protein bodies that ranged from 1.08 to 1.13 at 20 days after anthesis [36].

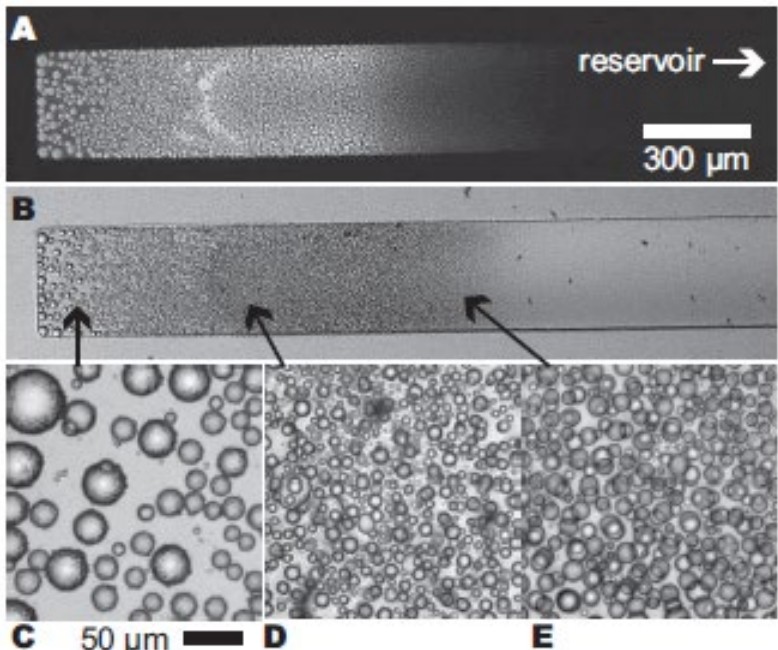

**Figure 2.** Microphotographs of the dense phases of γ-gliadins induced by steady accumulation in pure water. 'Large-scale' view of the accumulation process and nucleation of a dense, liquid phase in fluorescence mode (**A**) and bright field mode (**B**). Details of the dense phase at several positions along the microchannel showing several states of aggregation: (**C**) rough individual droplets of concentrated protein solution in the most concentrated zone, (**D**) smaller individual droplets in the intermediate zone, (**E**) droplets of concentrated protein solution sitting on the top (glass) and bottom (PDMS) surfaces of the microchannel.

*3.2. Raman Scattering in the Dense Phase of γ-Gliadins*

For the Raman scattering study, a dense phase of bare γ-gliadins in water was generated by microevaporation, and the reservoir was flushed with pure water in order to confine a finite amount of proteins in the microsystem (Figure 1B). Upon this specific kinetic pathway, the protein droplets tended to coalesce and formed an inhomogeneous system, which was studied using confocal Raman microspectroscopy. We used a confocal setup to minimize the contribution to the Raman scattering from the walls of the microsystem, actually made of glass for the top side and PDMS for the bottom side. The contribution of PDMS to the Raman scattering was significantly diminished by closing the confocal pinhole but was still present, despite a confocal depth of field ($\approx 5$ μm), much finer than the height of the microsystem (50 μm). We however kept this contribution and did not try to subtract the background. Indeed, the confocal volume changes when changing the refractive index of the sample and thus any background subtraction is inappropriate although not impossible in principle. We first collected Raman spectrum of dehydrated γ-gliadins in order to obtain a reference (Figure 3, bottom), and we used the amide I band to normalize all spectra. On this spectrum, we identified most of the molecular vibrations of the protein [37] amide I (1659 cm$^{-1}$) and III (1250 cm$^{-1}$), CH (1450 cm$^{-1}$), and CN from proline residue [38] (1450 cm$^{-1}$) vibration modes. The position of the amide I band suggested secondary structures within γ-gliadins were mainly composed of unordered and α-helices structures. This observation is in accordance with the previous spectroscopic studies of γ-gliadins, which indicated the presence of α-helices in the unrepeated domain and β-reverse turns

and PPII-helices in the repeated domain [8,12]. Within the microdevice, we studied two different regions: a concentrated zone of proteins droplets (Figure 3 middle), and a less dense region between these droplets, a zone which nevertheless displayed some Raman signal (Figure 3 top). The spectra collected were much less intense than spectra from the dried protein, a consequence of lower concentrations in particular for the diluted region. In the dense zone (Figure 3, middle), intense bands were attributed to phenylalanine (619, 1005, and 1607 cm$^{-1}$), tryptophan (878 cm$^{-1}$), and scissoring of the CH bonds combined with the stretching of the CN bonds in prolines (1451 cm$^{-1}$).

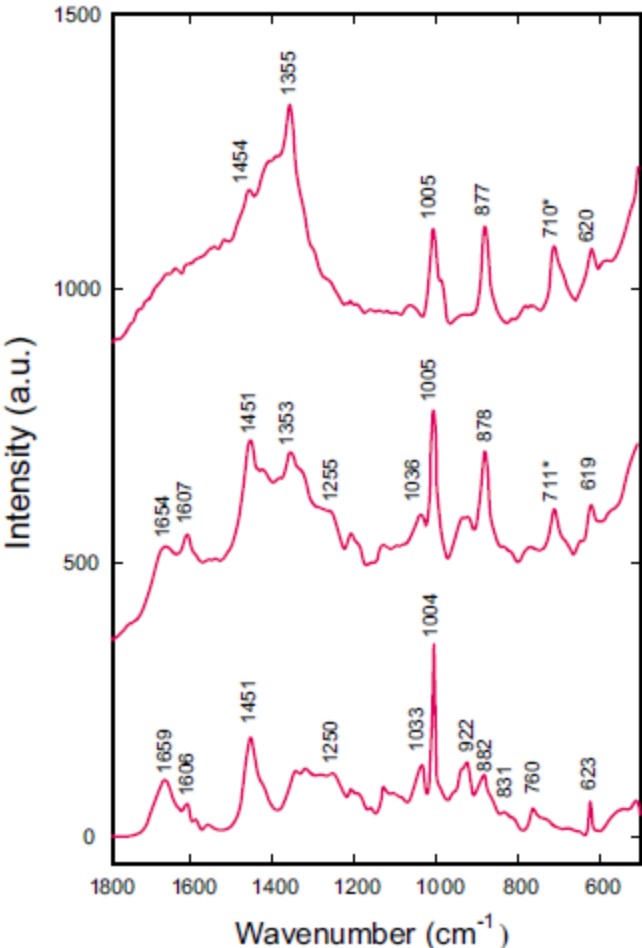

**Figure 3.** γ-Gliadin Raman spectra. (bottom) Lyophilised powder of the pure protein, (middle) dense zone of γ-gliadin droplets, and (top) dilute region between droplets. All spectra were normalized by the intensity of the amide I band. The star (*) indicates the band (around 710 cm$^{-1}$) ascribed to the PDMS constituting the microfluidic device; this band, even if small, is always present, despite the confocal mode of the microscope.

Finally, an intense band (1355 cm$^{-1}$) existing for the protein in both the concentrated and diluted zones (not in the dehydrated form) was attributed to the tryptophan residues in a non-homogeneous environment [39]. In the diluted zone between droplets (Figure 3, top), far less bands were visible, and the intensity ratio between the various bands was noticeably different. In particular, the tryptophan band exhibited an intensity that strongly depended on the environment, and it was evidenced here that the protein was therefore differently organized in the dense and diluted zones. In contrast, the phenylalanine bands did not depend on solvent effects but rather on the orientation of the residues. Here, the two modes at 1005 and 620 cm$^{-1}$ corresponded to vibration modes that were roughly perpendicular [40] the 1005 cm$^{-1}$ mode was assigned to the aromatic ring breathing, whereas the 620 cm$^{-1}$ mode was assigned to the ring bending. Their intensity ratio was similar in the dense zone

(4.6) and in the dried protein (4.3), whereas it was much less important in the diluted zone (2.8). With the interaction between the incident beam and the Raman signal being maximal for Raman tensor oriented parallel to the polarization of the incident beam [41], the aromatic rings of the phenylalanine residues should be preferentially oriented perpendicular to the PDMS surfaces in the diluted zone. In the concentrated zone, phenylalanine residues should be predominantly isotropic such as in the dried protein. Similarly, the very low intensities of amide I and 1450 cm$^{-1}$ bands in the diluted zone could be ascribed to specific orientations of the Raman tensors. The C=O and CN bonds in proline residues would be preferentially oriented perpendicular to the interface. The evolution of these intensity ratios suggested a specific orientation of γ-gliadin side chains relative to the interfaces in the diluted zones but not in the dense regions.

## 4. Discussion

Our optical observations demonstrated the occurrence of a liquid–liquid phase separation of γ-gliadins during the microevaporation process, and would indicate a short-range attractive potential. This potential would be the result of various interactions between proteins: Van der Waals, H-bonds, and hydrophobic and electrostatic interactions. In the particular case of γ-gliadins, the low effective charge value of γ-gliadins at neutral pH (+2.4 at pH 7), considered to be at the origin of the low aqueous solubility of this protein family, should not induce strong repulsive electrostatic potentials. Moreover, hydrogen bonding should be one of the main driving forces leading to a strong attractive potential and therefore protein–protein association due to the presence of numerous glutamine residues in its polypeptidic chain (≈35%). Different models were developed to compute attractive systems: the sticky hard sphere model potential of Baxter [42–44] or the screened coulomb interaction potential of Yukawa [45,46]. Supplementary quantitative results would, however, be required in our case to decipher between these models. The liquid–liquid phase separation can operate at constant potential, but using our device, the protein potential could be modified during the accumulation process: the electrostatic repulsive forces between proteins could be screened due to the increase of residual salt concentration present in the initial protein solution. Indeed, very low concentrations of salt into the initial solution can result in very high salt concentration into the system, as all species are concentrated during the process. Moreover, the heterogeneous nucleation of the dense phase observed on PDMS surfaces suggested a heterogeneous nucleation-growth mechanism.

The spectroscopic signature of oriented phenylalanine residues measured in the diluted phase can be rationalized, considering that during the heterogeneous nucleation, proteins adsorb onto hydrophobic interfaces with a specific orientation of aromatic residues such as phenylalanine. The signal measured for diluted phase would therefore be dominated by the proteins adsorbed at interfaces to the detriment of diluted proteins in bulk. This strong adsorption at interfaces is in accordance with our previous studies performed at the air–water interface [28,30,31]. In these previous studies, we proposed that the flat-oriented repetitive domain at interfaces would act as an anchor for subsequent protein–protein interactions. The present results are in agreement with this result as phenylalanine residues, which are specifically oriented and are mostly located in the repetitive part of the γ-gliadin sequence [47]. However, this specific orientation would not be maintained in the bulk dense phase.

These experiments also showed that γ-gliadins could exist as a non-precipitated form in aqueous environments—very diluted and concentrated liquid phases were actually observed. This result constitutes a strong argument to explain the mobility of gliadins measured in a biological context [48]. Finally, the microevaporator can be seen as a very simple biomimetic model of the endoplasmic reticulum (ER) where gliadins are gradually synthesized with an increasing rate along maturation [6]. This system allows for the accumulation of proteins in a confined geometry such as in the ER but does not include membranes. We previously showed that there was an interaction between gliadins and model membranes [28,30,31], but the interaction appeared non-specific. According to these

results, interfaces formed by membranes into the biological context would only serve as nucleation points for the formation of dense phases. This membrane could therefore be mechanically destabilized by the protein adsorption [49] and membrane budding due to proteins accumulation would occur, leading to the formation of proteins bodies.

## 5. Conclusions

The microevaporator is particularly well suited for the study of protein phase diagrams since very low quantities are necessary and high protein concentrations can be obtained. In our case, the gradual concentration of γ-gliadin aqueous solutions in the microevaporator enabled a liquid–liquid phase separation to be highlighted with coexistence of dense and diluted phases. Observations at the optical scale indicated that the dense phase nucleated at interfaces and grew to form droplets in bulk. These droplets did not provide evidence of any specific organization (optically isotropic with crossed polarizers, Raman spectra similar to the lyophilized protein). However, a specific orientation of phenylalanine residues in the diluted regions, observed by Raman spectroscopy, supported the hypothesis that proteins were firstly adsorbed at PDMS interfaces with a specific orientation. Nevertheless, mechanisms at molecular scale remain to be elucidated. Moreover, the observation of rough surfaces in the droplets of the dense phase is unexplained, leading to the conclusion that a better understanding of the dense phase microstructure is required. From a biological point of view, these experiments can be regarded as a mimic of the gradual protein synthesis and assembly in the confined environment constituted by the endoplasmic reticulum lumen, leading to the formation of protein bodies. However, it is worth noting that the concentration of salts in the microevaporator is a real limit of the system. The observation of a dense liquid phase obtained by the increase of γ-gliadin concentration in aqueous media supports an endoplasmic reticulum retention mechanism mediated by the intrinsic physicochemical properties of the wheat storage proteins.

**Author Contributions:** Conceptualization, J.L. and D.R.; methodology, A.B. and L.N.; validation, J.L., L.N. and D.R.; formal analysis, A.B.; investigation, A.B.; writing—original draft preparation, A.B.; writing—review and editing, J.L., L.N. and D.R.; supervision, L.N. and D.R.; funding acquisition, J.L. and D.R. All authors have read and agreed to the published version of the manuscript.

**Funding:** We acknowledge financial support from INRA and CNRS within the framework of the Groupement de Recherches "Assemblages de Macromolécules Végétales". Jacques Leng thanks Région Aquitaine for support and funding.

**Data Availability Statement:** Not applicable.

**Acknowledgments:** We are grateful to Jean-Pierre Compoint and Dominique Melcion for the purification of the gliadins. Finally, we acknowledge Jean-Baptiste Salmon for his involvement in the microfluidic part of this project.

**Conflicts of Interest:** The authors declare no conflict of interest.

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
