# Peer review of "Dense Phases of γ-Gliadins in Confined Geometries"

_colloids, doi:10.3390/colloids5040051_

Round 1

Reviewer 1 Report

The authors investigated the liquid-liquid phase separation of γ-Gliadins solution in a channel between PDMS and glass by using a microfluidic device. γ-Gliadins riched droplets are observed in the bulk as well as at the interfaces. The droplets formed at the interfaces are caused by the heterogeneous nucleation due to the short range interaction between liquid and interfaces. The results are interesting to the readers of Colloids and Interfaces. I would recommend for publication after a minor revision.

The authors claimed that the droplets are formed at the interfaces of PDMS and glass substrate. However,  it is not clear which interface is mainly responsible for the heterogeneous nucleation and at which interface droplet occurs earlier. This may be addressed if the wettabilities of γ-Gliadins droplet on the glass and PDMS membrane are known. I am wondering if the wettabilities of the γ-Gliadins droplet on these two interfaces are known.  Also, it is not clear to me why droplets detach from the interfaces to generate bulk-droplets. Is there a phase separation in the bulk to form droplets? Another mechanism might be the surface-directed phase separation.

I believe the heterogeneous nucleation may be explained by the Gibbs adsorption mechanism and Cahn's wetting transition theory. Can the authors comment on this?

Reviewer 2 Report

Banc et al. reported a microfluidic system for studying the binary phase diagram of gliadin. With the microfluidic device, authors observed that gliadin formed nucleation site on the surface first, and then grew into the bulk solution. I recommend the publication of this work after addressing the following concerns.

  1. Introduction should be separated into several paragraphs.
  2. The concentration gradient needs to be described quantitatively. How much difference in concentrations between the concentrated region and the diluted region?
  3. What is the role of surface contributing to the nucleation process? Is there any difference between PDMS surface and glass surface?
  4. For the Raman experiments, what is focal depth? From figure 1, it seemed that the bottom surface is glass. Please clarify.
  5. P2 line 51 and line 54 there are typos in the sentences.
  6. P3 line 72, 73, 74 and 76, there are typos in the sentences.

Round 2

Reviewer 2 Report

Publish as is.